# Digitizing mass spectrometry data to explore the chemical diversity and distribution of marine cyanobacteria and algae

Tal Luzzatto-Knaan[1]*[†], Neha Garg[1][†], Mingxun Wang[2], Evgenia Glukhov[3], Yao Peng[4], Gail Ackermann[5], Amnon Amir[5], Brendan M Duggan[1], Sergey Ryazanov[6], Lena Gerwick[3], Rob Knight[5], Theodore Alexandrov[1,6], Nuno Bandeira[1,2], William H Gerwick[1,3]*, Pieter C Dorrestein[1,2,3]*

[1]Collaborative Mass Spectrometry Innovation Center, Skaggs School of Pharmacy and Pharmaceutical Sciences, University of California San Diego, San Diego, United States; [2]Center for Computational Mass Spectrometry and Department of Computer Science and Engineering, University of California San Diego, San Diego, United States; [3]Center for Marine Biotechnology and Biomedicine, Scripps Institution of Oceanography, University of California San Diego, San Diego, United States; [4]Department of Chemistry and Biochemistry, University of California San Diego, San Diego, United States; [5]Departments of Pediatrics and Computer Science and Engineering, University of California San Diego, San Diego, United States; [6]European Molecular Biology Laboratory, Heidelberg, Germany

*For correspondence: tal. luzzatto@mail.huji.ac.il (TL-K); wgerwick@ucsd.edu (WHG); pdorrestein@ucsd.edu (PCD)

[†]These authors contributed equally to this work

Competing interests: The authors declare that no competing interests exist.

**Abstract** Natural product screening programs have uncovered molecules from diverse natural sources with various biological activities and unique structures. However, much is yet underexplored and additional information is hidden in these exceptional collections. We applied untargeted mass spectrometry approaches to capture the chemical space and dispersal patterns of metabolites from an in-house library of marine cyanobacterial and algal collections. Remarkably, 86% of the metabolomics signals detected were not found in other available datasets of similar nature, supporting the hypothesis that marine cyanobacteria and algae possess distinctive metabolomes. The data were plotted onto a world map representing eight major sampling sites, and revealed potential geographic locations with high chemical diversity. We demonstrate the use of these inventories as a tool to explore the diversity and distribution of natural products. Finally, we utilized this tool to guide the isolation of a new cyclic lipopeptide, yuvalamide A, from a marine cyanobacterium.

## Introduction

The marine environment is extraordinarily rich in both biological and chemical diversity. It is estimated that nearly 90% of all organisms living on earth inhabit marine and coastal environments (*Blunt et al., 2012*; *Kiuru et al., 2014*). Moreover, marine organisms are recognized as an extremely rich source of novel chemical entities that have potential utility in medicine, personal health care, cosmetics, biotechnology and agriculture (*Gerwick and Moore, 2012*; *Kiuru et al., 2014*; *La Barre, 2014*; *Rastogi and Sinha, 2009*). Marine cyanobacteria ('blue-green algae') are considered the most ancient group of oxygenic photosynthetic organisms, having appeared on earth some three billion

**eLife digest** Cyanobacteria and algae are found in all oceans around the globe. Like plants, they can use sunlight as a source of energy in a process called photosynthesis. As a result, these organisms are important sources of oxygen and another vital nutrient called nitrogen for other marine organisms. Many of these organisms also produce a variety of other chemicals known as "natural products" to help them to survive in their environments. Some of these natural products have shown potential as medicinal drugs. The search for new chemicals with useful medicinal properties has led researchers to collect samples of algae and cyanobacteria from various locations around the world.

An approach called mass spectrometry is often used to identify new chemicals because it can provide information about the structure of a molecule based on how much its fragments weigh. Luzzatto-Knaan et al. used mass spectrometry to search for new chemicals in samples of algae and cyanobacteria that had been collected by diving and snorkeling in a wide variety of tropical marine environments over several decades.

The experiments reveal that the organisms in these samples produce a diverse range of chemicals, most of which were previously unknown and have not been found in other similar environmental collections. The data were grouped together into eight major collection areas covering different parts of the tropics. The samples from some areas contained a wider variety of chemicals than others. Within each collection area, some molecules were found to be very common whereas others were only present at specific locations. To highlight the distribution of these natural products, Luzzatto-Knaan et al. display the data on a world map.

Further experiments used this approach as a guide to extract a previously unknown chemical called yuvalamide A from a marine cyanobacterium. The next challenge would be to associate the geographical patterns of chemicals to their potential ecological roles. This approach offers a new way to explore large-scale collections of environmental samples to discover and study new natural products.

years ago. It is thought that their extensive evolutionary history explains, at least in part, the extensive nature of their structurally-unique and biologically-active natural products (*Burja et al., 2001*; *Chlipala et al., 2011*; *Schopf, 2012*; *Whitton and Potts, 2012*). Indeed, since the 1970s, several hundred unique chemical entities have been discovered from marine cyanobacteria, and these have been found to possess a remarkable spectrum of biological activities including anticancer, anti-inflammatory, antibacterial, anti-parasitic, neuromodulatory, and antiviral, among others (*Blunt et al., 2015*; *Nunnery et al., 2010*; *Tan, 2007*; *Villa and Gerwick, 2010*). Examples of marine cyanobacterial natural products that are under investigation for anticancer applications include the cyclodepsipeptides apratoxin A and F, the nitrogen-containing lipid curacin A and the lipopeptides dolastatin 10 and carmaphycin B (*Bai et al., 1990*; *Blokhin et al., 1995*; *Luesch et al., 2001a*, *2001b*; *Pereira et al., 2012*; *Tidgewell et al., 2010*). These natural product templates have inspired the synthesis of lead compounds using medicinal chemistry approaches, and have resulted in one clinically approved drug (Brentuximab vedotin) for the treatment of cancer; several others are in various stages of clinical and preclinical evaluation (*Gerwick and Moore, 2012*; *Luesch et al., 2001a*; *2001b*; *Tan et al., 2010*; *Tidgewell et al., 2010*; *Uzair et al., 2012*). Unfortunately, such fruitful natural product materials, rich with so much unexplored potential, oftentimes disappear upon completion of academic careers and closing of laboratories.

In this study, we used a mass spectrometry based approach to digitize (convert to data format that can be stored, shared and analyzed by computational tools) the chemical inventory of an established marine cyanobacteria and algae collection in order to better evaluate its diversity and probe for novel natural products. Additionally, while a number of biologically potent natural products have been isolated from marine cyanobacteria and algae, the overall molecular profile of these organisms has not been compared to other classes of bacteria or terrestrial/freshwater cyanobacteria (*Chlipala et al., 2011*). To address this, we analyzed the crude extracts of a relatively large number of algae and cyanobacteria, as well as their derived chromatographic fractions, by high resolution

liquid chromatography tandem mass spectrometry (HR-LC-MS/MS). We then compared this metabolomics dataset to four existing public datasets [Mass spectrometry Interactive Virtual Environment (MassIVE) repository] within the Global Natural Product Social (GNPS) platform (*Wang et al., 2016*).

The marine samples used in this study were derived from over 300 field collections from locations in the Indian, Indo-Pacific, Central and South Pacific, and Western Atlantic (Caribbean) oceans with initial field taxonomic identifications as predominantly cyanobacteria and macro-algae. It should be noted that these are environmental samples, and therefore are inherently communities of organisms and are not single axenic cultures. Additionally, three samples were from non-axenic laboratory cultures of the cyanobacteria *Phormidium* (No.1646) and *Lyngbya* (No. 1933, 1963); the latter two were recently classified as *Moorea* (*Supplementary file 1*) (*Engene et al., 2012*). From both a chemical and biosynthetic perspective, these three cultures represent the most intensively studied organisms in this dataset (*Engene et al., 2012*; *Kleigrewe et al., 2015*; *Mevers et al., 2014*; *Pereira et al., 2010*; *Williamson et al., 2002*). While these and other of our cultured filamentous cyanobacteria cannot yet be grown axenically, their MS-derived molecular data represent marine cyanobacterial metabolomics markers that aid in data analysis. Furthermore, in exploring the chemical diversity of marine cyanobacterial and algal assemblages, we established a cartographic platform that combines LC-MS data and geographic locations that facilitates the discovery of new chemical scaffolds as well as identifies geographical areas of high chemical diversity (*Boeuf and Kornprobst, 2009*). The discovery of such 'hotspots' may reveal new patterns of phylogenetic diversity as well as identify geographical areas with enhanced bioprospecting opportunities.

Assessment of the chemical diversity of these marine cyanobacterial and algal collections involved four major steps: (1) **Collection** - including permits, field collection, transport, extraction, fractionation and metadata recording; (2) **Data acquisition and digitization in public repository** - by LC-HRMS/MS to generate molecular fingerprints; (3) **Data analysis and visualization** - clustering similarly structured compounds as molecular families within the GNPS platform, identification of known molecules and assessing the richness and diversity between as well as within samples; (4) **Discovery** - identification of geographical patterns of distribution, differentiating common from regiospecific natural products, dereplication of new derivatives and the discovery of previously uncharacterized natural products.

## Results

Over the past 30 years, a significant number of cyanobacterial and algal collections were obtained under the appropriate governmental permits using scuba diving and snorkeling in the Caribbean (Puerto Rico, Grenada, Panama), Central and South Pacific (Hawaii, Palmyra Atoll, French Polynesia, Fiji), Indo-Pacific (Indonesia, Papua New Guinea), and Indian Oceans (Madagascar, South Africa). These collections represent natural assemblages of benthic filamentous marine cyanobacteria and various classes of macro-algae (Rhodophyta, Chlorophyta and Phaeophyceae). Each collection was extracted ($CH_2Cl_2$/MeOH, 2:1) and fractionated using a standardized vacuum liquid chromatography protocol (*Supplementary file 1*). Approximately 2600 fractions originating from 317 marine collections, including the unfractionated crude samples, were analyzed by reversed phase ultra-performance liquid chromatography (RP-UPLC) coupled with high resolution quadrupole time-of-flight mass spectrometry (HR-qTOF-MS) to obtain retention times and MS and tandem MS/MS fragmentation spectra (LC-MS/MS). Nearly 6000 spectra were collected for each LC-MS/MS run, generating in excess of 15.6 million spectra for the samples in this study.

These data were analyzed using the GNPS platform enabling the extensive organization of the LC-MS/MS data (*Wang et al., 2016*). GNPS detected features (i.e. molecules) based on MS/MS spectra and MS intensities. Further, some of these MS/MS spectra were identified by matching to spectral libraries available on GNPS. Even in the absence of MS/MS spectral matching to known reference MS/MS spectra, GNPS molecular networking can associate structurally related molecules that exhibit similar MS/MS fragmentation patterns into molecular families (*Watrous et al., 2012*; *Yang et al., 2013*).

# Comparative metabolomics and chemical diversity of large scale datasets

Few tools allow assessment of the chemical diversity within large and diverse MS datasets such as those in the current study (*Bouslimani et al., 2014*; *Charlop-Powers et al., 2015*; *Luzzatto-Knaan et al., 2015*; *Purves et al., 2016*). To determine whether this collection of marine cyanobacterial and algal communities possessed unique chemical diversity, the LC-MS/MS data were analyzed with publicly available data sets accessed via the GNPS-MassIVE database (For datasets see methods) (*Wang et al., 2016*). These datasets include LC-MS/MS data from terrestrial and marine actinobacteria (1000 samples), lichens (132 samples), marine sponges and corals (260 samples) and freshwater cyanobacteria (535 samples). Collection of all datasets was acquired using tandem mass spectrometry to obtain unique fragmentation fingerprints for each detected molecule.

To evaluate the chemical diversity of these source materials, we employed diverse MS-based informatic approaches, including multivariate analyses and molecular networking. Multivariate approaches, such as principal component analysis (PCA) and partial least squares-discriminant analysis, (PLS-DA), have been widely employed to assess the chemical diversity of metabolomics data (*Worley and Powers, 2013*). Here, for comprehensive metabolomics analysis, we applied principal coordinate analysis (PCoA), a statistical method that provides an overview of the dissimilarities between samples based on the metabolome of an individual sample relative to all other samples in the analysis (*He et al., 2015*) The relatedness between samples is calculated by the dissimilarity distance matrix and displayed in a three-dimensional plot where each sphere represents a sample with specific MS/MS features. We utilized the Bray-Curtis distance matrix applied to all MS/MS features, as exported from GNPS, using the QIIME platform (*Caporaso et al., 2010*). The separation of the samples in PCoA space highlights the differences between the various origins of the samples (*Figure 1A*). The lichen samples separate from the marine associated samples, although they overlap the spatial distributions of some cyanobacteria populations, perhaps because most lichens contain cyanobacteria, bacteria and fungi (*Mushegian et al., 2011*). Interestingly, among the five environmental samples, the marine cyanobacteria/algae and freshwater cyanobacteria are more similar in their metabolomics profiles than any other of the source materials, even though the two aquatic environments are quite distinct (*Figure 1A*). Nevertheless, while presenting the highest similarity compared to the other datasets, the marine cyanobacteria/algae and freshwater cyanobacteria samples show a clear differentiation in their chemical inventories (*Figure 1B*). To quantify the common and differential chemical features driving these distances, we generated a Venn diagram using the MS/MS features. (*Figure 1C*,). This analysis revealed that only 13.7% of the marine cyanobacteria/algae molecular features overlapped with other datasets, highlighting that 86.3% were features unique to these collections. Among the molecular commonalities, a number of the mass spectrometry signals were identified as matches to reference spectra in GNPS. These included a mixture of primary metabolites such as amino acids, fatty acids, structural lipids, pheophytin and pheophorbide A (chlorophyll derived products), as well as common mass spectrometry background signals such as formate clusters, plasticizers and polymers. Remarkably, despite being most similar by PCoA analysis, no reference MS/MS spectra of the known freshwater cyanobacterial natural products matched with our marine cyanobacteria and algal collection dataset.

In exploring the chemical diversity of the marine cyanobacteria and algae communities, we hypothesized that either the organism collected (cyanobacterial or algal) or the geographical location was responsible for sample-to-sample metabolomics differences (*Figure 2*). Our PCoA analysis did not indicate differentiation based on field annotations as algal versus cyanobacterial (*Figure 2A*), and the non-significant clustering based on geographical origin, highlights the vast chemical diversity of samples within the same region (*Figure 2B*). PCA, the most common dimensionality reduction and visualization method used for metabolomics analysis, displayed a parallel outcome with minor trends being observed (*Supplementary file 1*, *Figure 2—figure supplement 1*).

Construction of fraction libraries has become a common strategy to make more efficient the discovery of bioactive molecules in natural products studies so as to reduce sample complexity for biological screening as well as increase the titer of minor constituents (*Bugni et al., 2008*; *Harvey et al., 2015*). In the current analysis, the fraction library yielded over five times the number of molecular features compared to the crude extracts, thereby enabling a more thorough assessment of chemical richness and diversity (*Figure 3A*). The fraction library resulted in 1094 molecular

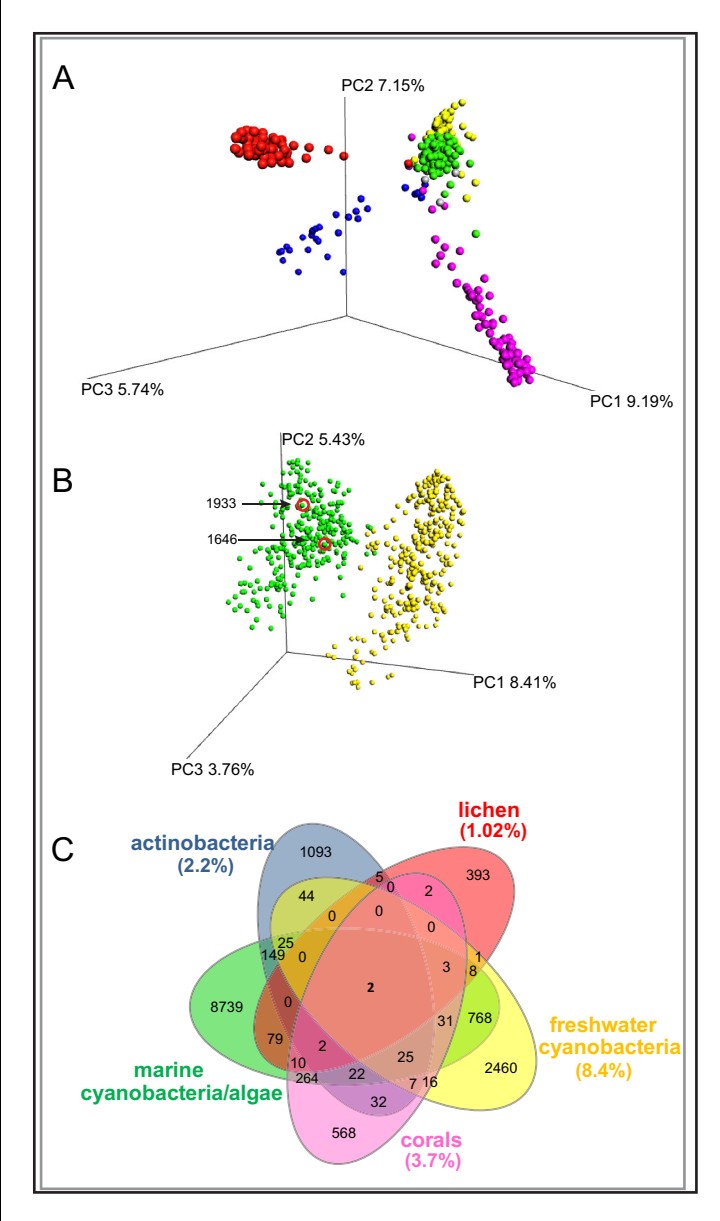

**Figure 1.** MS/MS features as generated in GNPS are shown for marine cyanobacteria and algae (green), marine and terrestrial actinobacteria (blue), lichens (red), freshwater cyanobacteria (yellow) and corals (pink). (**A**) PCoA plots of 300 samples randomly selected from each dataset display the distance between samples based on molecular features using Bray-Curtis dissimilarity matrix. (**B**) Marine cyanobacteria and algae and freshwater cyanobacteria. Laboratory cultures of *Phormidium* 1646, *Lyngbya* 1933 are highlighted in red circles. Each sphere represents the full sample metabolome (**C**) Venn diagram display of overlapping MS/MS features. Percentage of overlapping features with the marine cyanobacteria and algae dataset are given in parenthesis in their respective colors (*Supplementary file 1*).

families compared to 132 in the crude extract analysis. This tenfold difference highlights the advantage of using fraction libraries for chemical and biological assay screens. By comparing the number of features as well as the number of underlying spectra for a given molecular family (*Nguyen et al., 2013*), we could identify particular chemical scaffolds that are more diverse in their structural variations and are more abundant in our collections. For example, the barbamide molecular family (*Figure 3B*) is comprised of 40 molecular features represented in 985 individual spectra and found in

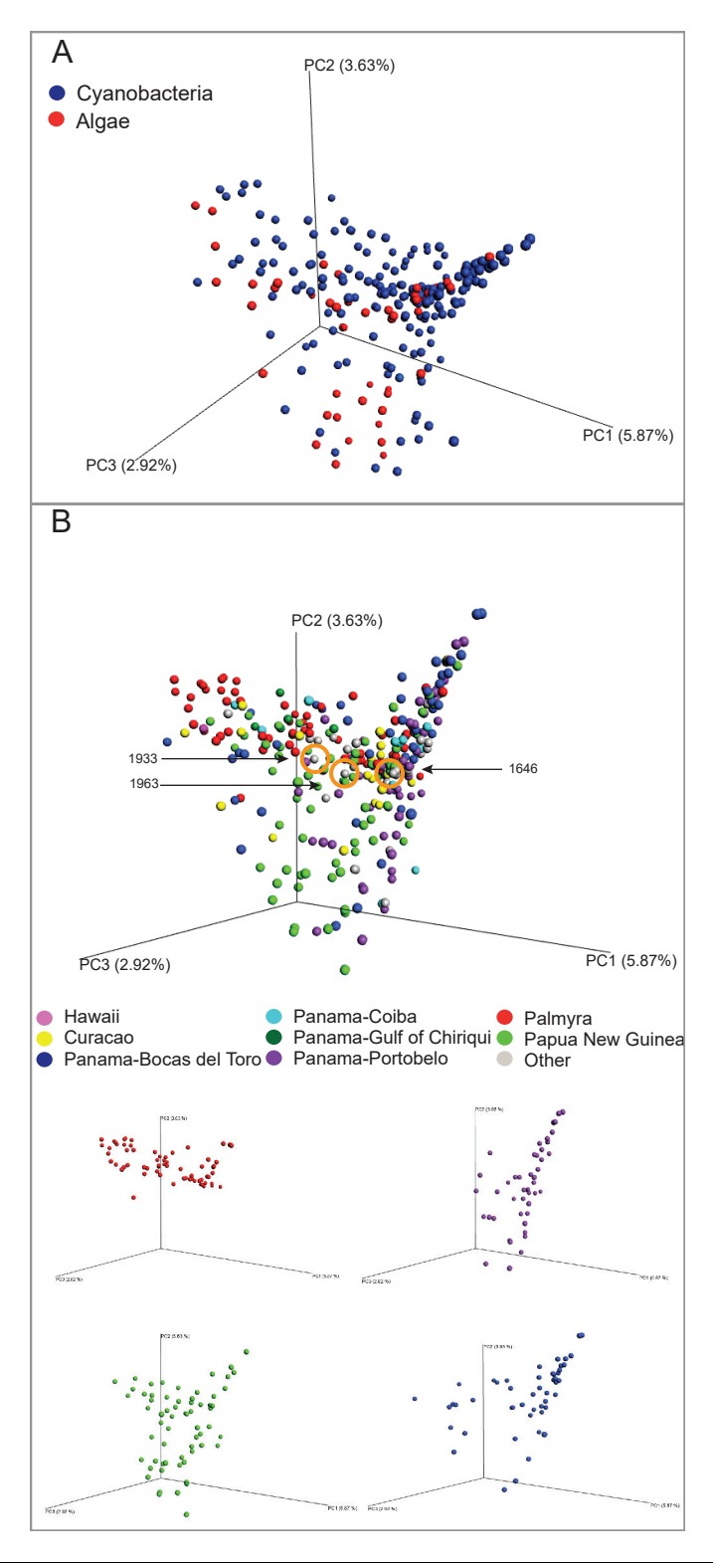

**Figure 2.** MS/MS feature diversity between and within sampling sites. PCoA plots of crude extract molecular features using Bray-Curtis, dissimilarity matrix. Each point represents a single sample and points are colored by metadata. (**A**) PCoA plot shows the distance between samples based on field identified classification. (**B**) PCoAs color coded by geographical origin shows the distance of all locations together (B: upper panel) and the individual
*Figure 2 continued on next page*

*Figure 2 continued*

PCoA plots of the four collection sites with most samples (B: lower panel). Laboratory cultures of *Phormidium* 1646, *Lyngbya* 1933 and *Lyngbya* 1963 are highlighted in orange circles.
The following figure supplement is available for figure 2:

**Figure supplement 1.** PCA plot of crude extracts of cyanobacteria and algae collections.

75 collections. By contrast, the palmyrolide A molecular family has only two associated features and consists of 32 spectra that are found in five collections (*Figure 3B*). Using a rarefaction analysis, both the crude extracts and the fraction library were found to level off under the current experimental conditions of analysis, indicating that the sample collections are saturating the chemical diversity of these materials (*Figure 3A*).

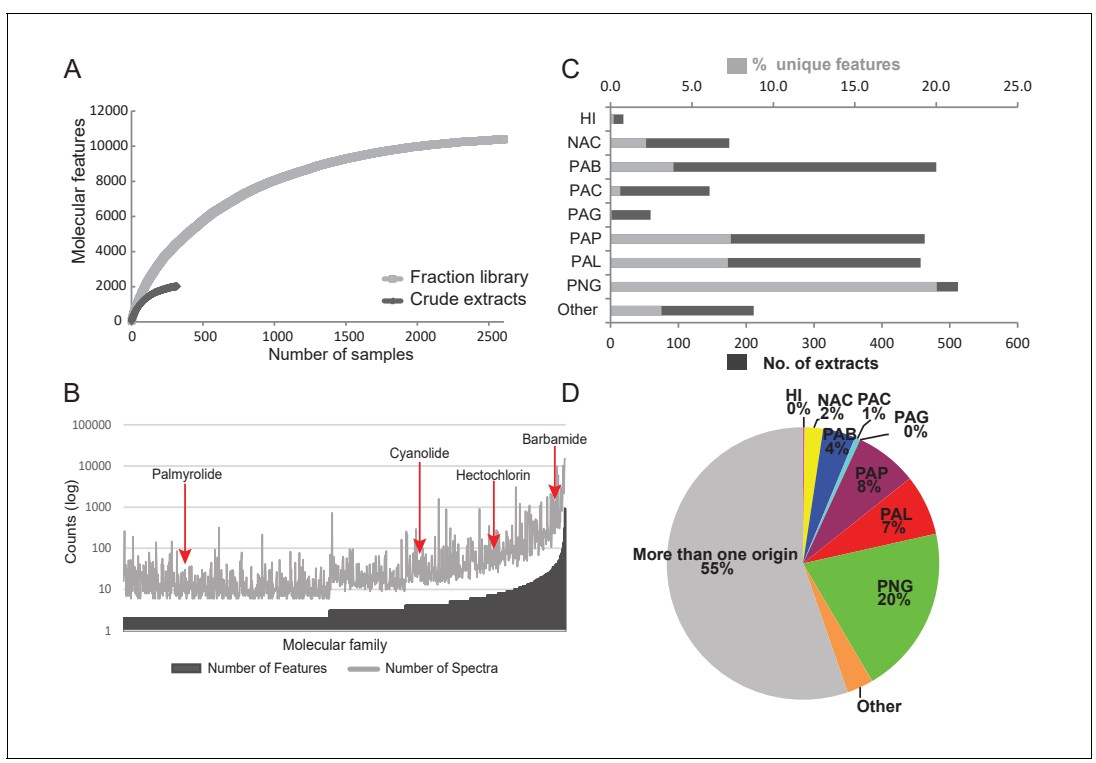

**Figure 3.** Chemodiversity and richness of molecular features based on MS/MS data. (**A**) Rarefaction curve of cyanobacteria collection library showing the chemical richness of crude extracts vs. the fraction library. (**B**) Abundance of molecular families: Bars represent the number of features and spectra comprising each molecular family clustered by GNPS molecular networking. (**C**) Bar chart depicting both the total number of extracts from a given location and the percent contribution of unique features to the entire dataset. (**D**) Pie chart representing the percentage of unique molecular features attributed to origin of the sample. HI-Hawaii, NAC-Curacao, PAB-Panama Bocas del Toro, PAC-Panama Coiba, PAG-Panama Gulf of Chiriqui, PAP-Panama Portobelo, PAL-Palmyra Atoll, PNG-Papua New Guinea.
The following figure supplements are available for figure 3:

**Figure supplement 1.** Molecular network of marine cyanobacterial natural products with annotated molecular families.

**Figure supplement 2.** Dereplication of the apratoxin molecular family.

**Figure supplement 3.** Dereplication and spatial distribution for new derivatives of the barbamide molecular family.

## Mapping the distribution of marine natural products

To identify locations of high chemical diversity among sampling sites represented in this study (locations with more unique natural products compared to other regions), we compared the MS/MS features found only in a single location to the overall diversity of the entire dataset (*Figure 3C and D*). To do this, we normalized the contribution of features from a single location to a percentage contribution as some locations were sampled more frequently than others based on metadata information collected across three decades (*Figure 3C*). For example, approximately 4% of all MS/MS features contributing to the overall diversity are from Panama-Bocas del Toro (PAB) whereas about 7–8% are each contributed from Panama-Portobelo (PAP) and Palmyra Atoll (PAL), and 20% are from Papua New Guinea (PNG) (*Figure 3D*).

To explore the chemical nature of differentially produced metabolites in the marine cyanobacterial and algal sample collections, GNPS based molecular networking was used and the network was visualized in Cytoscape (For molecular network and Cytoscape file see Materials and methods, *Figure 3—figure supplement 1*) (*Purves et al., 2016*; *Shannon et al., 2003*; *Watrous et al., 2012*). As MS/MS spectral alignment detects molecular features, a set of structurally related features connected by edges is termed a 'molecular family'. Experimental MS/MS spectra were dereplicated (also described as 'finding known unknowns' by the metabolomics community) against the community contributed mass spectral 'GNPS library' and third party spectral libraries including MassBank, HMDB, ReSpect and NIST 2014 housed within GNPS analysis infrastructure (*Horai et al., 2010*; *Johnson and Lange, 2015*; *Sawada et al., 2012*; *Wang et al., 2016*; *Wishart et al., 2013*). This is currently the largest molecular network generated and only possible through the GNPS interface (*Wang et al., 2016*). From the more than 15.6 million spectra collected, 6249 MS/MS spectra were dereplicated to existing publicly accessible reference data, with 91 of the matches to previously characterized cyanobacterial and algal derived natural products from 30 molecular families (*Nguyen et al., 2013*). The spectral library matches from the marine cyanobacteria/algae collection included: 74 matches to the GNPS library, 7 matches to the NIH pharmacologically active compounds library, 3 matches to the NIH natural product library, 1 match to the NIH clinical collection, 1 match to the FDA natural products library and 5 matches to the Faulkner legacy library. This is a match rate of about 0.04% which is much lower than database matches of the average untargeted metabolomics experiments (1.8%, [*Wang et al., 2016*]) and metabolomics analyses of model samples such as *E. coli*, human cell lines, mice or humans (*da Silva et al., 2015*). The low match rate to existing natural product libraries within GNPS highlights that these marine cyanobacterial/algal communities are significantly underexplored from a chemical perspective.

The hundreds of unannotated molecular families from these samples suggest that they contain a significant number of natural products that remain uncharacterized. Molecular networking allows putative structural propagation through a molecular family because most molecules of similar structure fragment similarly. By investigating selected molecular families, we identified previously characterized metabolites and previously undescribed derivatives (*Supplementary file 2* and *Figure 3—figure supplements 2 and 3*). For example, we identified the spectral cluster representing the apratoxin molecular family by a spectral match to the apratoxin B library standard. Further dereplication of this molecular family using literature information and manual analysis of MS/MS spectra revealed several known apratoxin derivatives (A, D, F, G, H) (*Figure 3—figure supplement 2*) (*Luesch et al., 2002*; *Tidgewell et al., 2010*; *Watrous et al., 2012*; *Yang et al., 2013*). Additional apratoxin derivatives are suggested on the basis of mass differences of common functional groups (*Figure 3—figure supplement 2*). Further, GNPS identified a molecular family comprised of the barbamides (*Orjala and Gerwick, 1996*). Within this molecular family, we were able to deduce several candidate derivatives based on fragmentation patterns and presence of chlorine isotopes (*Figure 3—figure supplement 1* and *Figure 3—figure supplement 3* [1]) (*Orjala and Gerwick, 1996*; *Yang et al., 2013*). For example, a 14 Da decrease from the y ion of barbamide most likely results from loss of the amide methyl group, leading to *N*-desmethyl barbamide (*Figure 3—figure supplement 3* [2]). *O*-desmethyl barbamide was previously published and is identifiable by the loss of 14 Da from the characteristic MS/MS fragment containing the three chlorine atoms (*m/z* 242.98) (*Kim et al., 2012*). The 34 Da reduction from the y ion appears to be associated with the substitution of the Phe group by Leu, a conservative hydrophobic replacement for A-domains that accept Phe in the NRPS-based biosynthesis of barbamide. Such a conservative substitution has previously been observed in other

NRPS biosynthetic systems (*Figure 3—figure supplement 3* [3]) (*Flatt et al., 2006*). This exchange of amino acids gives rise to a new molecular entity; however, it is reminiscent of related compounds, such as dysidin, which was isolated from the cyanobacteria-harboring sponge *Dysidea herbacea* (*Ilardi and Zakarian, 2011*).

Because a majority (55%, *Figure 3D*) of molecules detected in this dataset were found in more than one sampling location, we explored the spatial distribution and abundance of specific natural products using binning networks. For this visualization, the abundance/presence of a metabolite was correlated to the eight collection sites. In the network, each node represents a distribution pattern of molecules, where the node size and color correspond to the relative number of features that share the same distribution (*Figure 4*). Of the 256 conceivable regiospecific patterns (based on the eight collection sites) some molecules share similar distribution patterns, while some distribution patterns are absent or very minor in their abundance (light colored nodes). To visualize the distribution of select molecules identified via molecular networking, in our eight major sampling sites, MS/MS feature intensity values were projected onto a 2D geographical map based on GPS coordinates using the open-source tool 'ili (an open-source tool for 2D and 3D data visualization created by our laboratories, https://github.com/ili-toolbox/ili [*Protsyuk et al., 2017*], available as a Google Chrome app; a copy is archived at https://github.com/elifesciences-publications/ili) (see legend of *Figure 4* for more information). For example, out of the 33,481 detected features, dolastatin 10 along with 72 other chemical features share the same widespread distribution pattern (Curaçao (NAC), PAB, Panama-Coiba (PAC), PAL, PAP, PNG), while palmyramide A and 3033 other chemical features are endemic to PAL, at least within the scope of this data set, and are not found in any of the other sampling locations. Additionally, while barbamide is widely distributed, putative analogs are encountered relatively infrequently (*Figure 4B* and *Figure 3—figure supplement 3*).

## Discovery of new natural products

To emphasize the application of MS-based geographic network analyses for the discovery of uncharacterized natural products, we examined several chemical features from locations of high chemical diversity, namely the PAP, PAL and PNG sites. One sample was chosen from PAP and targeted for the isolation of an *m/z* 536.333 molecule from a cluster with no described family members (For molecular network see Materials and methods, molecular family #483). This specific molecular feature was observed in several cyanobacterial collections of *Lyngbya* from PAP obtained in different years from 2007 to 2013 (collections in Dec 2007, Sept 2010 and Jan 2013). The MS/MS data indicated that this compound was peptidic and contained the amino acids glycine, valine (or *N*-methyl glycine), leucine or isoleucine, and 2-hydroxyvaleric acid (Hiv). The presence of this latter residue suggested it was the product of a non-ribosomal peptide synthetase (NRPS) (*Kehr et al., 2011*). An additional residue was observed corresponding to the fragment mass *m/z* 149.094, compatible with the formula $C_{10}H_{13}O^+$. Further analysis of the sequential fragment losses suggested a putative structure of [Gly-Ile/Leu-Hiv-Val-$C_{10}H_{13}O]^+$. A search for a putative peptide with a molecular mass *m/z* 536.333 in various chemical databases (*Blunt et al., 2015*, *Blunt et al., 2012*; *Luzzatto-Knaan et al., 2015*; http://pubs.rsc.org/marinlit/) was unsuccessful, suggesting that this natural product was uncharacterized.

In order to confirm the mass spectrometry-based partial structural assignment, we isolated and structurally characterized this natural product of mass *m/z* 536.333 (*Figure 5*, *Figure 5—figure supplements 1* and *2*, *Supplementary file 3* [4]). Isolation of compound **4**, given the common name of 'yuvalamide A', was guided by LC-MS analyses. NMR analysis (DMSO-$d_6$) of the pure compound confirmed the presence of Gly, Val, Ile, and hydroxyisovaleric acid, and revealed the presence of 2,2-dimethyl-3-hydroxy-7-octynoic (Dhoya) as the $C_{10}H_{13}O^+$ fragment (*Figure 5*, *Figure 5—figure supplements 1* and *2*, *Supplementary file 3*). These NMR-based determinations correlate well with the structural predictions obtained by mass spectrometry.

The other chemical entities in the yuvalamide A molecular family were present at very low abundance, preventing their isolation and structural analysis by NMR. However, based on the structure of yuvalamide A and its fragmentation pattern by LC-ESI-MS/MS, we deduced putative structures of these co-metabolites. An analog of yuvalamide A (*Figure 5—figure supplement 3* compound **5**, yuvalamide B, [M+H]$^+$ *m/z* 538.3487, [M+Na]$^+$ *m/z* 560.3320) was detected by HRMS. Its fragmentation was nearly identical to compound **4**, differing only in the fragment belonging to Dhoya, suggesting this to be the alkene analog 2,2-dimethyl-3-hydroxy-7-octenoic acid (Dhoea). Co-occurring

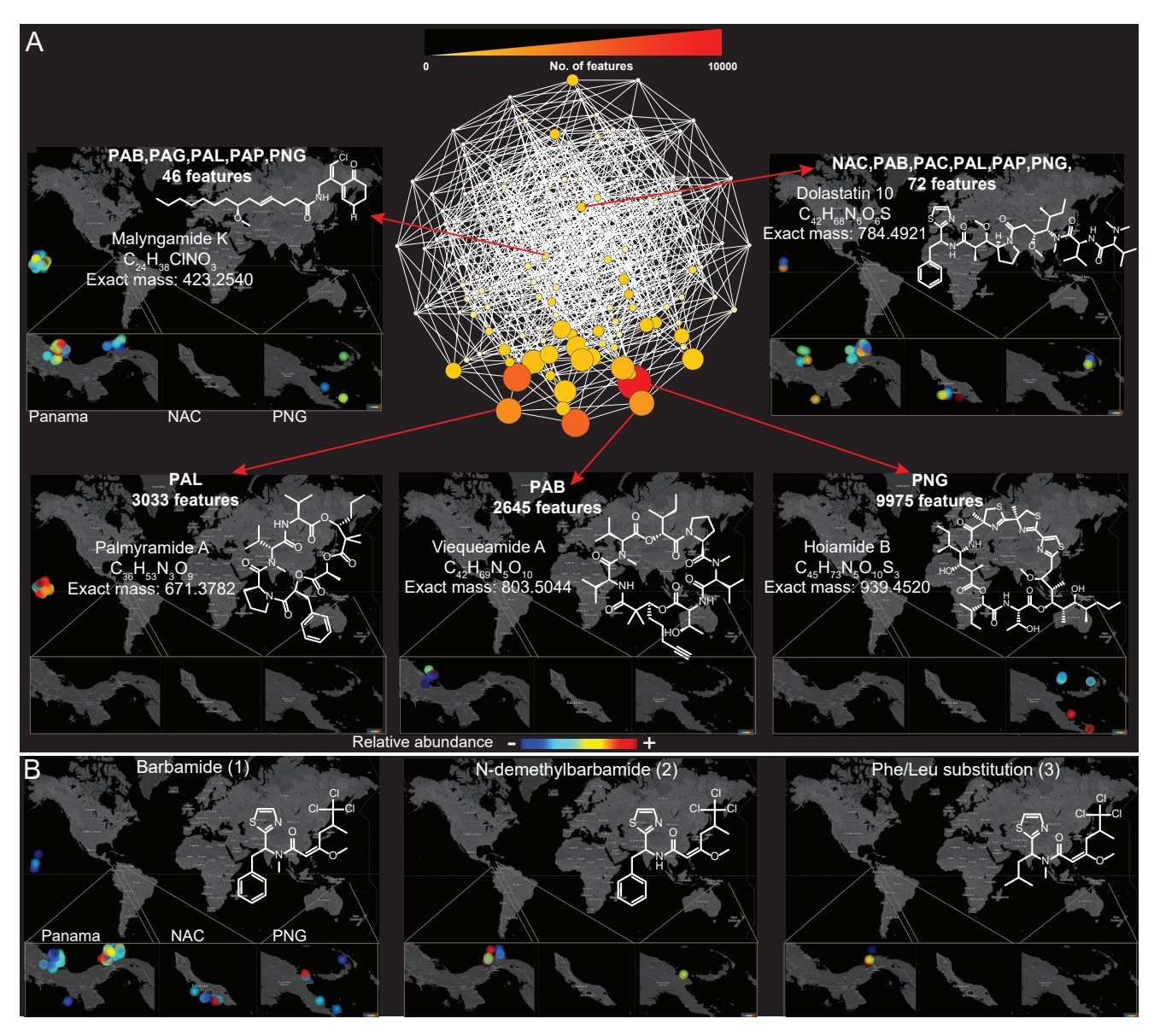

**Figure 4.** Spatial maps showing the geographic distribution of selected cyanobacterial natural products. (**A**) Features were binned by their distribution patterns across the eight main geographical locations, each bin represented by a node with edges linking bins differing by one location. Number of features in each bin is presented according to size and color scale from white (0) to red (10,000) as indicated by the scale bar on the top. Spatial patterns are represented for selected natural products within these bins. Inserts in each map display zoomed-in sections of Panama, Curaçao and Papua New Guinea. Each sample is designated to a specific coordinate based on GPS coordinates of the collection site (multiple samples are represented by spots placed around the collection site). See the URL provided below. (**B**) Spatial maps display chemogeographical distribution and abundance of barbamide (**1**) and two barbamide analogs (**2**, **3**). Relative abundance is presented by Jet color scale from low (blue) to high (red). HI-Hawaii, NAC-Curaçao, PAB-Panama Bocas del Toro, PAC-Panama Coiba, PAG-Panama Gulf of Chiriqui, PAP-Panama Portobelo, PAL-Palmyra Atoll, PNG-Papua New Guinea.

(To use the open source tool 'ili please open the following link in Google Chrome: http://ili-toolbox.github.io/?cyano/bg.png;cyano/intensities.csv. Please wait until the data is loaded and visualized, then click on the Mapping submenu, and change the scale to Logarithmic, and Color map to Jet. For flipping through maps corresponding to different molecules, please click on the name of the molecule shown above the colorbar and select another molecule either with a click or with up-down arrow keys. Alternatively, you can refer to the example tab and choose cyanobacteria natural products. If you experience any problems, please contact theodore.alexandrov@embl.de.)

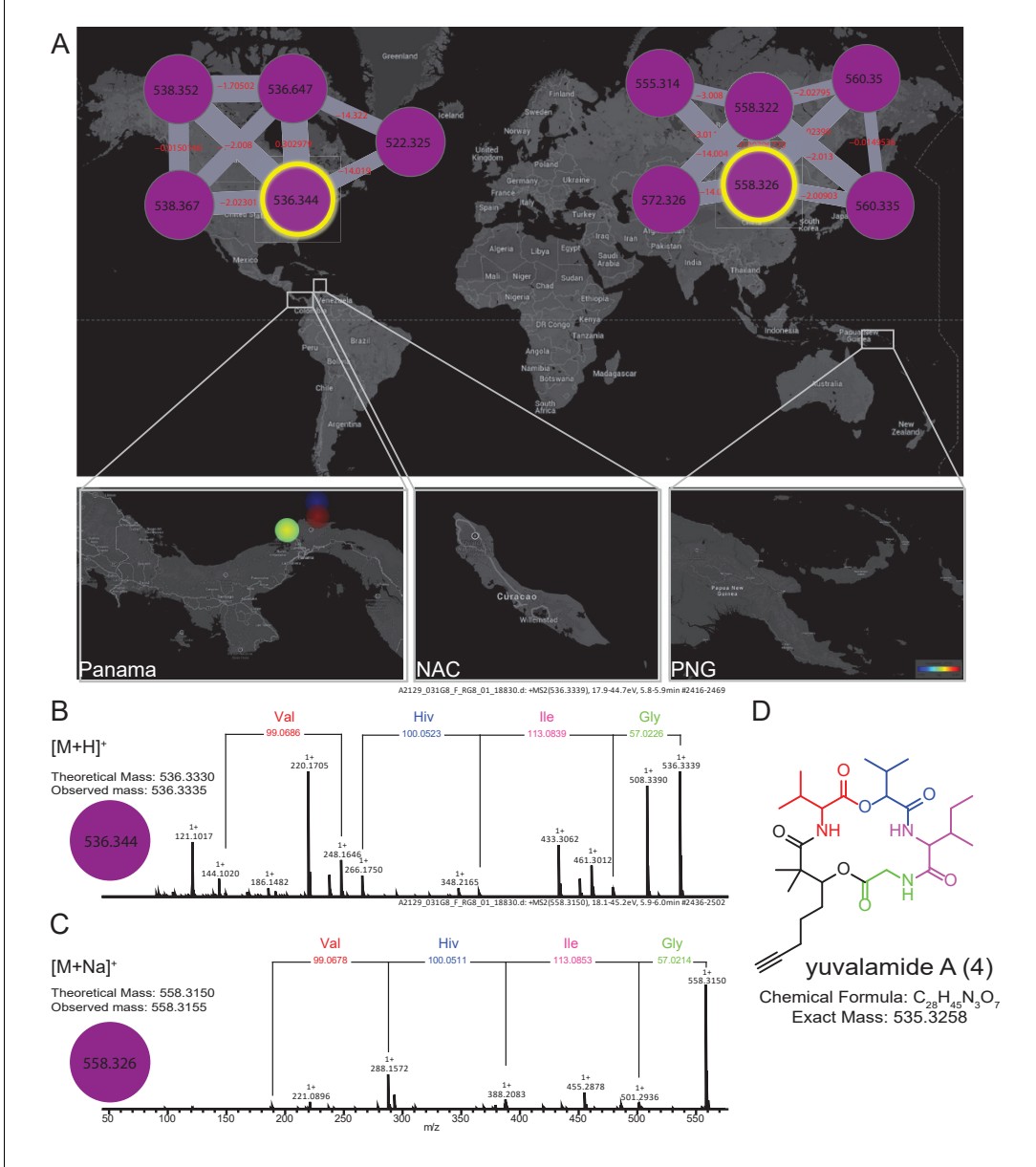

**Figure 5.** The molecular family and structure elucidation of a novel natural product yuvalamide A (4), isolated from a Panama-Portobelo (PAP) cyanobacterial collection. (**A**) Molecular families and the spatial distribution of yuvalamide A [M+H]+ and [M+Na]+ ions are highlighted within yellow circles. MS/MS spectra display the linear structure for the non-ribosomal peptide (NRP) fragments Gly-Ile-Hiv-Val. (**B**) for the [M+H]$^{+}$ and the (**C**) [M +Na]$^{+}$ ions. (**D**) The full elucidated structure of yuvalamide A (4) as confirmed by NMR analysis (*Figure 5—figure supplements 1* and *2*, *Supplementary file 3*).

The following figure supplements are available for figure 5:

**Figure supplement 1.** Structure identification of yuvalamide A.

**Figure supplement 2.** MS/MS fragmentation of yuvalamide A.

**Figure supplement 3.** Dereplication of yuvalamide molecular family.

analogs possessing the Dhoya or Dhoea unit have been observed previously in marine cyanobacterial natural extracts. The Dhoya subunit is usually the more abundant of the two, as observed in this case for yuvalamide A (*Boudreau et al., 2012*; *Han et al., 2005*, *2011*; Luesch et al., 2001; *Sitachitta et al., 2000*; *Wan and Erickson, 2001*). Moreover, the geographical distribution of yuvalamide B is the same as yuvalamide A (*Figure 5*), supporting the proposal that these share a common biosynthetic origin. Two additional yuvalamide analogs were observed and tentative structures were assigned based on MS (*Figure 5—figure supplement 3*).

## Discussion

Over the last 40 years, a number of academic natural product discovery programs have generated unique and significant collections from a variety of sources. However, the samples as well as the associated collection and characterization information is often buried in individual laboratory materials and documentation, and frequently this disappears with academic turnovers. To explore a new paradigm for characterizing and preserving such information over a longer time period for the community at large, we harnessed recent advances and metabolomics approaches to deeply annotate the metabolomes of these collections. We then evaluated this rich dataset along with collection information for its diversity, distribution and discovery of new natural products.

To date, biodiversity studies have mostly relied on available genomic information (*Bowen et al., 2013*; *Passarini et al., 2015*). While organisms might be phylogenetically related, their metabolomic repertoire may vary greatly. The detection of biosynthetic gene clusters indicates natural product potential, and has been successfully employed to predict new molecules, especially of the core structure (*Medema and Fischbach, 2015*; *Medema et al., 2015*; *Owen et al., 2015*; *Walsh, 2015*). However, post assembly modifications (e.g. halogenations, oxidations, methylations) are often times difficult to predict without the structures in hand, and therefore genomic analyses do not provide a complete assessment of the secondary metabolomes. Chemical diversity represents the expression of that genetic potential and offers a new strategy to explore differences and to evaluate the diversity encrypted in complex samples and various environments. Herein, we broadly inventoried the molecules that are present in a large library of field collected marine algae and cyanobacteria using LC-MS/MS. Using the mass spectrometry detectable natural products from each sample, the chemodiversity and the regional patterns of these samples were characterized.

It is reported in the literature that marine cyanobacteria are an exceptional source for diverse new natural products (*Blunt et al., 2015*; *Burja et al., 2001*; *Engene et al., 2011*; *Kiuru et al., 2014*; *Tan, 2007*). However, this perception is difficult to assess quantitatively in comparison to other marine or terrestrial life forms. To gain insights into how unique are the secondary metabolites of marine cyanobacteria and algae, we compared these MS data with other metabolomic data sets that are available from the public domain.

The mass spectrometry data from marine cyanobacteria/algae collections obtained over the past 30 years indicate that, at a global level, this group is rich in metabolite features that do not overlap with other available metabolomic data sets (*Figure 1C*). Common multivariate analyses used for metabolomics were successful in distinguishing the different datasets based on their unique metabolomic inventories.

Key to a chemical pattern analysis is the ability to organize molecules on the basis of structural features along with their associated metadata attributes. GNPS molecular networking generates molecular features that are based upon fragmentation similarity, thereby providing resolution at the individual molecular level. This approach also enables the rapid matching of spectra to known molecules by comparison with various natural product MS/MS libraries (*Nguyen et al., 2016*; *Wang et al., 2016*; *Yang et al., 2013*). In this regard, the MS/MS spectrum of a molecule is a unique molecular signature or barcode. When structurally similar molecules are fragmented, the resulting MS/MS spectra are very often quite similar because they contain similarly positioned labile bonds (*Wang et al., 2016*). Using molecular networking, a visual representation of these 'barcodes' is produced by grouping similar patterns. Molecular networking enables the evaluation of the MS/MS based detectable chemical space, and groups molecules that have similar biosynthetic origins and resultant structures. At the organism level (e.g. invertebrates, plants, corals), marine environments such as the Papua New Guinea archipelago (PNG) are considered to be biodiversity 'hotspots' (*Bowen et al., 2013*). However, not only is PNG a biodiversity hotspot, molecular networking

quantitatively revealed that this region is rich in the presence of unique natural products compared to other sampled locations in this study (*Figure 3*) (*Bowen et al., 2013*; *La Barre, 2014*). Although this is demonstrated here for these marine algal and cyanobacterial collections, it would be interesting to evaluate if high biodiversity always correlates with high chemical diversity, and as a corollary, how chemical diversity changes upon a shift in biodiversity. The geographical patterns of molecule distribution could be further investigated via the tools presented here, given the appropriate sampling strategy. While molecular networking revealed the locations with lowest and highest chemical diversity, visualizing these data on a world map enabled assessment of the distribution patterns of specific molecules within the eight sampled regions. This analysis also enabled targeting samples with higher abundance of molecules of interest, and may therefore assist in characterizing exceptionally high-producers of natural products. We have demonstrated that some chemical entities were found in common to multiple locations (e.g. barbamide, dolastatin 10) whereas others appeared to be unique and endemic to specific sampling sites (e.g. hoiamide B, palmyramide) (*Figure 4*). This observation is supported by site resampling over several years, an insight gained as a distinctive attribute of this unique collection of marine samples.

Additional tracing of the metadata associated with a given molecular family can assist in identification of the producing organism, and facilitate future biosynthetic and ecological investigations (*Coates et al., 2014*; *Harvey et al., 2015*; *Kleigrewe et al., 2015*; *Leal et al., 2016*; *Owen et al., 2013*). Finally, we demonstrated the utility of this overall approach through the discovery and subsequent structural elucidation and geographical distribution of yuvalamide A (**4**), reported here for the first time.

## Conclusion

The integration of molecular networking with geographical mapping revealed that marine algae and cyanobacteria have widely varying metabolite distributions. As we can see in this dataset most molecules are broadly distributed in disparate physical locations, whereas others are highly specific to particular locations. Whether this specificity to a given location is due to unique species with distinct metabolic capabilities, or to specific environmental factors regulating expression of these compounds, is unknown at present. Moreover, field collections are a complex mixture of microorganisms that hold a unique repertoire of natural products that may not be discoverable under laboratory conditions, even with sequenced strains. Therefore, our approach is uniquely capable of organizing the vast chemical diversity present in these complex environmental samples. Further, this approach could be used to identify locations with a higher level of chemical diversity as well as distribution patterns of natural products, potentially giving insights into their ecological roles. In turn, these perceptions may facilitate the search for novel natural products of utility to biomedicine and biotechnology.

## Materials and methods

### Collection and culturing

A total of 317 diverse marine cyanobacterial and benthic algae collections were hand-collected in Panama, Papua New Guinea, Hawaii, Madagascar, Palmyra Atoll, Curaçao, and a few other tropical marine sites at depths from 0.3 to 20 m with the aid of snorkel or scuba gear (*Supplementary file 1*). GPS location, preliminary field taxonomic identification of the specimens, and depth of collection were noted for most collections. Samples for subsequent chemical extraction were strained through a mesh bag to remove excess seawater, preserved in equal volumes of seawater and ethanol, transported to the laboratory, and stored at −4°C or −20°C until workup. Live samples of marine cyanobacteria were brought back to the laboratory in vented tissue culture flasks with 0.2 μm filtered native seawater and subsequently isolated to mono-cultures. These were cultured in SWBG-11 media with 35 g/L Instant Ocean (Aquarium Systems Inc.). Mono-cultures were grown at 28°C in a 16 hr light/8 hr dark cycle with a light intensity of ~7 μmol photon/s/m$^2$ provided by 40 W cool white fluorescent lights. Specific collection information is described in the Supporting Information.

### Extraction library

Chemistry samples were extracted repetitively with $CH_2Cl_2$:MeOH 2:1, dried *in vacuo*, and in most cases, fractionated into nine fractions (A-I) by silica gel vacuum liquid chromatography (VLC) using a stepwise gradient of hexanes/EtOAc and EtOAc/MeOH. Each fraction and crude extract was re-suspended at 5 mg/mL in pure DMSO and stored in 96-well plates at −20°C until analysis.

### UHPLC-MS/MS analysis

The extracted metabolites were analyzed with an UltiMate 3000 UHPLC system (Thermo Scientific) using a Kinetex 1.7 μm C18 reversed phase UHPLC column (50 × 2.1 mm) and Maxis Impact Q-TOF mass spectrometer (Bruker Daltonics) equipped with an ESI source. The data were acquired in positive ionization mode for parent mass ($MS^1$) and tandem MS for molecular fragmentation ($MS^2$). Amount of material injected from each sample was 3.3 μg. The data were acquired in positive ionization mode for parent mass ($MS^1$) and tandem MS for molecular fragmentation ($MS^2$). The gradient employed for chromatographic separation was 5% solvent B (ACN/$H_2O$/formic acid 98%/2%/0.1%) with solvent A ($H_2O$/ACN/formic acid 98%/2%/0.1%) for 1.5 min, a step gradient of 5% to 50% B in 0.5 min, held at 50% B for 2 min, a second gradient of 50–100% B in 6 min, held at 100% B for 0.5 min, 100–5% B in 0.5 min and kept at 5% B for 0.5 min at a flow rate of 0.5 mL/min throughout the run. The MS analysis was performed on a Maxis QTOF mass spectrometer (Bruker Daltonics), controlled by the Otof Control and Hystar software packages (Bruker Daltonics), and equipped with ESI source. MS spectra were acquired as previously described (*Garg et al., 2015*).

### Molecular networking

A molecular network was created using the online workflow at GNPS (*Wang et al., 2016*). To denoise, the data was filtered by removing all MS/MS peaks within ±17 Da of the precursor *m/z*. MS/MS spectra were filtered by choosing only the top six peaks in the ±50 Da window throughout the spectrum. The data were clustered with MS-Cluster with a parent mass tolerance of 1.0 Da and a MS/MS fragment ion tolerance of 0.5 Da to create consensus spectra. Consensus spectra that contained less than three spectra were discarded. A network was then created where edges were filtered to have a cosine score above 0.6 and more than four matched peaks. Further edges between two nodes were kept in the network if and only if each of the nodes appeared in each other's respective top 10 most similar nodes. The spectra in the network were then searched against GNPS spectral libraries. The library spectra were filtered in the same manner as the input data. The networking output was visualized using Cytoscape 2.8.3 free software (*Shannon et al., 2003*) and organized using the FM3 force directed layout (*Hachul and Junger, 2004*). Follow the link to the GNPS molecular network of multiple datasets: http://gnps.ucsd.edu/ProteoSAFe/status.jsp?task=e69c25e3fbcf4c09b6e0c75633565f95. Links to the cyanobacteria/algae datasets: ftp://massive.ucsd.edu/MSV000078568, ftp://massive.ucsd.edu/MSV000078892 and the corresponding network on GNPS: http://gnps.ucsd.edu/ProteoSAFe/status.jsp?task=0837888b3dab43efa5bf9a50254c7c8f. For the Cytoscape file ftp://massive.ucsd.edu/MSV000078568/other/.

### Statistics and diversity analysis

The MS/MS data tables generated by GNPS for all analyzed extracts and fractions was converted to a BIOM table (http://biom-format.org/) (*McDonald et al., 2012*). Each of the tables was then used for calculation the sample-sample distance metrics using binary metric (Binary-Jaccard) using QIIME (*Caporaso et al., 2010*) version 1.9. Principal coordinates were calculated and visualized based on various metadata categories (region, organism etc.) using EMPeror (*Vázquez-Baeza et al., 2013*). Rarefaction curves of 20 random iterations were generated by GNPS.

### Spatial mapping of mass spectrometry data

Spatial molecular maps were created with background map from a Google Map screenshot with several regions of interest enlarged and plotted as insets to the overview map. By visual examination, we assigned to each sampling spot the (x,y) pixel coordinates on the background map based on GPS coordinates taken at the time of collection. For multiple samples from the same location, spots were placed in adjacent locations. For each node of the molecular network generated in GNPS, we considered the area under the curve of all mapped spots, scaled them from 0% to 100% and

assigned colors to all intensities by using the 'jet' color map, color-coding low intensities with blue, high intensities red, and other intensities with a colored gradient between blue and red. The rendering of the maps was performed by the 'ili tool for 2D and 3D spatial mapping (https://github.com/ili-toolbox/ili). The color was made gradually disappearing with highest intensity at the center of the sampling location and almost no intensity at the boundary of the spot. For this, a logarithmic combination of the default color and assigned color was used, with the coefficient exponentially depending on the distance from the sampling spot center.

### Structure elucidation of yuvalamide A (4)

A preliminary structure was deduced based on a HRMS ($C_{28}H_{45}N_3O_7$, 8° unsaturation) and HRMS/MS (Bruker Daltonics Maxis Impact qTOF) fragment analysis. Exact masses observed were [M+H]$^+$ $m/z$ 536.3335 and [M+Na]$^+$ $m/z$ 558.3155 corresponding to a molecular formula of $C_{28}H_{45}N_3O_7$ (<0.9 ppm). Neutral losses of glycine, leucine/isoleucine and valine were observed. An additional loss matching the formula of 2-hydroxyisovaleric acid (Hiv) was observed suggesting yuvalamide was a non-ribosomal peptide. This MS-based fragment analysis suggested a Gly-Ile/Leu-Hiv-Val sequence with an additional $C_{10}H_{13}O$ fragment. The peptide appeared to be cyclic based on degrees of unsaturation, and by database query, was a new compound.

Analysis of 2D NMR data (gCOSY, TOCSY, HSQC, HMBC) confirmed three of the five substructures (Val, Hiv, Gly) and clarified the structures of the remaining two (*Figure 5—figure supplement 1*, *Supplementary file 3*). The Ile/Leu substructure was identified as Ile from COSY correlations between the beta protons at 1.96 ppm and a methyl group at δ 0.86, the beta protons at δ 1.96 and a pair of germinal protons at δ 1.49 and 1.13, and between the geminal protons to a second methyl group at δ 0.81. Finally, a 2,2-dimethyl-3-hydroxyoctynoic acid (Dhoya) residue was similarly identified. The *gem*-dimethyl arrangement was revealed by key HMBC correlations from CH$_3$-9/10$_{Dhoya}$ to the carbonyl carbon C-1$_{Dhoya}$, quaternary carbon C-2$_{Dhoya}$, and oxygenated methine CH-3$_{Dhoya}$. This latter resonance was connected by COSY to a linear sequence of three methylene groups, the terminal methylene of which was long range coupled to alkynyl carbons (δ$_C$ 71.7 and 84.5), one of which (δ$_C$ 84.5) showed a HMBC correlation to a terminal alkynyl proton (δ$_H$ 2.78). The assembly of these substructures into an overall cyclic depsipeptide structure was enabled by a key HMBC correlation between Dhoya H3 (δ$_H$ 4.71) and the Gly carbonyl (δ$_C$ 168.6) (*Figure 5—figure supplement 1/2*, *Supplementary file 3*). Additional HMBC correlations linking all the substructures verified the MS/MS-deduced sequence described above (*Figure 5—figure supplements 1* and *2*, *Supplementray file 3*).

## Acknowledgements

The authors would like to thank Dr. Gabriel Navarro for his help with the NMR structure elucidation and Dr. Vanessa V Phelan for her valuable insights on the manuscript. TLK was supported, in part, by BARD, the United States – Israel Binational Agricultural Research and Development Fund, Vaadia-BARD Postdoctoral Fellowship Award no. FI-494–13. This work was supported by NIH grant GM107550 to PCD, LG and WHG and European Union FP7 and H2020 programmes grants No. 305259 and 634402. The authors declare no conflict of interest.

## Additional information

### Funding

| Funder | Grant reference number | Author |
|---|---|---|
| National Institutes of Health | GM107550 | William H Gerwick |
| European Commission | FP7 | Theodore Alexandrov |
| H2020 | 305259 | Theodore Alexandrov |
| Vaadia-BARD Fellowship | No.FI-494-13 | Tal Luzzatto-Knaan |
| H2020 | 634402 | Theodore Alexandrov |

The funders had no role in study design, data collection and interpretation, or the decision to submit the work for publication.

## Author contributions

TL-K, Conceptualization, Data curation, Formal analysis, Investigation, Methodology, Writing—original draft, Project administration, Writing—review and editing; NG, Data curation, Formal analysis, Methodology, Writing—review and editing; MW, Software; EG, Methodology, Project administration; YP, Methodology; GA, Formal analysis, Methodology, Project administration; AA, Formal analysis, Validation, Project administration; BMD, Formal analysis, Validation, Methodology; SR, Software, Visualization; LG, Resources, Funding acquisition, Methodology; RK, Software, Validation, Writing—review and editing; TA, Software, Visualization, Methodology; NB, Software, Validation, Visualization; WHG, Conceptualization, Resources, Supervision, Funding acquisition, Methodology, Writing—review and editing; PCD, Conceptualization, Resources, Supervision, Funding acquisition, Writing—original draft, Writing—review and editing

## Author ORCIDs

Tal Luzzatto-Knaan, http://orcid.org/0000-0002-8392-0501

# Additional files

## Supplementary files

• Supplementary file 1. Samples list and metadata. Sample list and metadata are freely available upon registration at http://qiita.microbio.me Study ID:10125. Also refers to *Figures 2*, *3*, *4* and *5*.

• Supplementary file 2. Supplemental table 1: GNPS molecular networking MS/MS based identification of molecules and molecular families – Refers to *Figures 3* and *4*.

• Supplementary file 3. 2D NMR spectroscopic data for amino acids residues of yuvalamide A. $^{13}$C and $^{1}$H chemical shifts were determined by HSQC and HMBC spectra. Hiv = 2 hydroxyisovaleric acid (600 MHz for $^{1}$H, 125 for $^{13}$C), the solvent (DMSO-d6) and the temperature (298K).

## Major datasets

The following datasets were generated:

| Author(s) | Year | Dataset title | Dataset URL | Database, license, and accessibility information |
|---|---|---|---|---|
| Tal Luzzatto-Knaan | 2015 | 150717_Marine_lichen_actino_freshW_coral | http://gnps.ucsd.edu/ProteoSAFe/status.jsp?task=e69c25e3fbcf4c09-b6e0c75633565f95 | Publicly available at Global Natural Products Social Molecular Networking website (http://gnps.ucsd.edu/) |
| Tal Luzzatto-Knaan | 2014 | 141021_BigData_06_MCS3_MMP4_p31 | http://gnps.ucsd.edu/ProteoSAFe/status.jsp?task=0837888b3dab43e-fa5bf9a50254c7c8f | Publicly available at Global Natural Products Social Molecular Networking website (http://gnps.ucsd.edu/) |
| Tal Luzzatto-Knaan, Neha Garg, Mingxun Wang, Evgenia Glukhov, Yao Peng, Gail Ackermann, Amnon Amir, Brendan M Duggan, Sergey Ryazanov, Lena Gerwick, Rob Knight, Theodore Alexandrov, Nuno | 2017 | GNPS_Cyanobacterial_collection_Plate31 | ftp://massive.ucsd.edu/MSV000078892 | Publicly available via the Mass spectrometry Interactive Virtual Environment (MassIVE) (accession no: MSV000078892) |

| | | | | |
|---|---|---|---|---|
| Tal Luzzatto-Knaan, Neha Garg, Mingxun Wang, Evgenia Glukhov, Yao Peng, Gail Ackermann, Amnon Amir, Brendan M Duggan, Sergey Ryazanov, Lena Gerwick, Rob Knight, Theodore Alexandrov, Nuno Bandeira, William H Gerwick, Pieter C Dorrestein | 2016 | GNPS_Dorrestein_Gerwick_Cyanobacteria | ftp://massive.ucsd.edu/ MSV000078568/other/ | Publicly available via the Mass spectrometry Interactive Virtual Environment (MassIVE) (accession no: MSV000078568) |

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
