## [Decision Letter]

Thank you for submitting your article "Digitizing mass spectrometry data to explore the chemical diversity and distribution of marine cyanobacteria and algae" for consideration by *eLife*. Your article has been reviewed by three peer reviewers, and the evaluation has been overseen by a Reviewing Editor and Ian Baldwin as the Senior Editor. The following individual involved in review of your submission has agreed to reveal his identity: Tim Bugni (Reviewer #3).

The reviewers have discussed the reviews with one another and the Reviewing Editor has drafted this decision to help you prepare a revised submission.

Summary:

This article by Luzzatto-Knaan et al. employs advances in MS/MS molecular networking developed in the Dorrestein lab to "capture the chemical space and dispersal patterns" of natural products from a set of samples from previous sample campaigns. The strength of the approach taken by the authors relies on the use of high-resolution MS/MS data to digitize the chemical inventory of the existing marine cyanobacteria and algae collection. This approach is combined with a systematic clustering, within the GNPS platform (https://gnps.ucsd.edu/), of the resulting molecular and fragmentation data according to compound families and in order to visualize the diversity, richness and geographical distribution of different classes of natural products.

Importantly, by placing non-targeted MS/MS structural data processing at the forefront, this study represents an important shift from previous natural screening procedures that are typically guided by low-to-medium-throughput molecule bioactivity screening procedures. This analysis also confirms that marine cyanobacterial/algal communities are significantly under-explored from a chemical perspective. As such, the vast amount of information rapidly parsed through the GNPS-based approach may guide the isolation of new groups of metabolites, like the newly described metabolite yuvalamide A or of derivatives of known groups of metabolites such as the barbamides analogs and yuvalamide B.

All three reviewers were generally positive about the work presented in this study. Especially, all agreed that the authors generated an impressive amount of data and presented elaborated classification and visualization approaches to systematically evaluate these data. As the GNPS database continues to grow, the types of analyses presented in this study will certainly be able to answer questions that were previously not tractable.

However, important concerns were raised regarding shortcomings in biological and geographical sampling design and how these hamper some of the conclusions drawn by the authors. Essential revisions are therefore needed in which conclusions most specifically with respect to geographical specificity should be toned down.

Essential revisions:

1) One major concern raised by the reviewers relates to the conclusions reached by the authors regarding geographical specificity. The observation of regiospecific clusters is interesting but since no targeted sampling covering the same phyla in the respective regions has been done, the clustering does not necessarily relate to the geographic location but may translate from sampling biases. Hence, a quantitative evaluation and discussion as performed in Figure 3 is problematic despite the effort for normalization. Also, plotting the data on a world map gives a nice picture but in the light of the small sample size (only 8 locations sampled) the suggested comprehensive global character of the study is not established. For all these reasons, generalizations in the text regarding the interpretation of regiospecific clusters are problematic and should be avoided during the revision: e.g. "To visualize the global distribution of select molecules" should rather read “to visualize the distribution […] in our 8 sampling sites ". Along with this, authors should remove the core conclusion that this study comprehensively visualizes molecules that are distributed based on geography. Instead, we recommend that authors highlight that, given the appropriate sampling strategy, the geographical importance of molecule distribution could be investigated via the tools presented.

2) Figure 2 do not clearly show regiospecific clustering. For example, the Papua New Guinea samples seem to be found throughout the figure. Since the figure is plotting dissimilarity relationships, it seems counterintuitive to use such a measure to understand regiospecific clustering. A similarity relationship might more clearly show regiospecific clustering. Also, while the collections of Panama are divided to three distinct locations the collections, in what I consider a larger area, Papua New Guinea is treated as one location. If one merges the diversities from Panama it gives the same percentage as PNG. Could the authors comment on this point, which echoes with the above recommendation to tone down conclusions about geographical specificity?

3) Another concern is related to the selection of organisms includes (quite arbitrary) benthic filamentous marine cyanobacteria and various classes of macro-algae (Rhodophyta, Chlorophyta and Phaeophyceae) predominantly collected from warm waters (Carribean, Indo-Pacific, Indian Ocean…). It is nice to see but no surprise that MS/MS features for marine cyanobacteria and algae are different from marine and terrestrial actinobacteria, lichens, freshwater cyanobacteria and corals. In relation with the previous points on geographic specificity, the authors should more explicitly acknowledge this biological bias in the text.

4) Along the same line, the authors state that "86.3% were features that were unique to these collections." In some way, the authors should provide a better method of capturing the current state of GNPS to provide a benchmark. While Figure 1 provides the number of molecular features, it would be useful to have a better understanding of the number of molecules present in GNPS at the time the comparison was made. Perhaps, this information is captured in Supplementary file 4. Looking at Figure 1, it appears that most of GNPS might be comprised of data from cyanobacteria. How many of those features are represented in the current study? This information will be important for the reader to understand what the comparison entails.

[Editors' note: further revisions were requested prior to acceptance, as described below.]

Thank you for resubmitting your work entitled "Digitizing mass spectrometry data to explore the chemical diversity and distribution of marine cyanobacteria and algae" for further consideration at *eLife*. Your revised article has been favorably evaluated by Ian Baldwin (Senior Editor) and the Reviewing Editor.

The revised manuscript is an improvement in terms of clarity of interpretation of the region-specific metabolome mining analysis in light of technically-understandable shortcomings. Also, we agree with the authors that the lack of a comprehensive global character does not contradict the advances and potential impact of the work. However, we felt that the following points needed further considerations before this manuscript can be accepted for publication:

1) First, the Abstract still conveys the general idea that the authors conducted a comprehensive regio-specific analysis while only 8 sampling sites were considered. As discussed in the first round of review, this is a technically understandable shortcoming which does not alter the validity of the MS/MS and data analysis approaches, but the fact that the study is limited to 8 sampling sites should be stated in the Abstract.

2) The lower panel of Figure 2 which is referred to as panel C in the third paragraph of the subsection “Comparative metabolomics and chemical diversity of large scale datasets” is not labelled accordingly in the figure. This minor labelling discrepancy should be corrected.

3) Point 3 raised during the review concerned the interpretation of the comparison of features derived from Panama and Papua New Guinea (PNG) samplings to the overall chemical diversity across sampling sites. Most notably, if one merges the explained chemical diversities from Panama, it results in the same percentage as PNG. Considering the sampling diversity and existing metadata, the authors in their response to this point do not agree with the suggestion of merging the chemical diversity analyses inferred for each of the Panama current location categories. This argument for not merging the Panama location categories is reasonable; however, we feel that it remains inappropriate to draw the particular strong conclusion: "these numbers suggest that even though the sample sizes were similar, the detected chemical diversity varies greatly with PNG possessing the greatest chemical richness". This final sentence of the paragraph could be removed without altering the overall description of the value of the analytical approach used to explain sampling site-specific chemical diversity.

4) Finally, since the term "digitizing" or "digitization" is used in the title as well as throughout the text as a cornerstone step in the authors' workflow, it may be helpful for readers that one or two sentences are added to precisely define this term from a technical standpoint. Readers from different backgrounds will have different expectations as to what is meant by "digitizing".

---

## [Author Response]

*Essential revisions:*

*1) One major concern raised by the reviewers relates to the conclusions reached by the authors regarding geographical specificity. The observation of regiospecific clusters is interesting but since no targeted sampling covering the same phyla in the respective regions has been done, the clustering does not necessarily relate to the geographic location but may translate from sampling biases. Hence, a quantitative evaluation and discussion as performed in Figure 3 is problematic despite the effort for normalization. Also, plotting the data on a world map gives a nice picture but in the light of the small sample size (only 8 locations sampled) the suggested comprehensive global character of the study is not established.*

We thank the reviewers for raising this issue and would like to emphasize the limitations, considerations and benefits of this study. The main purpose on this work was to utilize a rare and unique collection of samples that has been collected for decades and with the abilities developed in our lab, to demonstrate the additional information that can be explored. Such unique collections exist in various laboratories that specializes in natural products studies that in many cases kept untouched for years or cleared out with academic turnovers. Seed banks and biodiversity collections also exist and may benefit from the potential this study is presenting as an example of longitudinal collections, where the technology is advancing far beyond the expectations when samples were actually collected or the purpose they were collected for. It is true that for this dataset we represent only 8 major locations, and we agree with the possibility of potential biases. However, these are the restrictions of the drug discovery effort that relies on authorizations and international permits for collecting biological material where the fate of the intellectual property outcome is clearly defined. As much as we would like to represent samples from the entire globe, we *do not* feel that the lack of it is contradicting the advances and potential this work is introducing. On the contrary, we highlight the publicly available tools developed in our labs (GNPS, QIIME and ili) to encourage their use and the sharing of information with labs that may have more comprehensive collections. The work presented and the analysis that is being made is illustrative of these tools; we feel that if we were not to present this analysis, the utility of these tools would not be apparent to the community.

*For all these reasons, generalizations in the text regarding the interpretation of regiospecific clusters are problematic and should be avoided during the revision: e.g. "To visualize the global distribution of select molecules" should rather read “to visualize the distribution […] in our 8 sampling sites".*

We appreciate and agree with the comment, and have changed the text according to reviewers’ suggestion as follows:

“To visualize the distribution of select molecules identified via molecular networking, in our 8 major sampling sites, MS/MS feature intensity values were projected onto a 2D geographical map based on GPS coordinates using the open-source tool ‘ili […]”

“While molecular networking revealed the locations with lowest and highest chemical diversity, visualizing these data on a world map enabled assessment of the distribution patterns of specific molecules within the eight sampled regions.”

*Along with this, authors should remove the core conclusion that this study comprehensively visualizes molecules that are distributed based on geography. Instead, we recommend that authors highlight that, given the appropriate sampling strategy, the geographical importance of molecule distribution could be investigated via the tools presented.*

We appreciate and agree with this important comment, and have added a note on this in the text as follows: “The geographical patterns of molecule distribution could be further investigated via the tools presented here, given the appropriate sampling strategy”.

*2) Figure 2 do not clearly show regiospecific clustering. For example, the Papua New Guinea samples seem to be found throughout the figure. Since the figure is plotting dissimilarity relationships, it seems counterintuitive to use such a measure to understand regiospecific clustering. A similarity relationship might more clearly show regiospecific clustering.*

We thank the reviewers for the comment and wish to clarify. The purpose of this analysis was to identify the *differentiating metadata factor* among these samples. Indeed, as mentioned in the text, based on the information in hand, the detected features of a single sample metabolome did not present a significant differentiation. Yet, it is important to mention that by looking at specific molecules, we do see regional specificities that we can account for in this dataset. To address reviewers’ concerns, we clarified the text as follows:

“Although our PCoA analysis did not indicate differentiation based on field annotations as algal versus cyanobacterial (Figure 2), some geographic origin and regiospecific clustering was observed in this analysis (Figure 2)”

Text was modified to: “Our PCoA analysis did not indicate differentiation based on field annotations as algal versus cyanobacterial (Figure 2), and the non-significant clustering based on geographical origin highlights the vast chemical diversity of samples within the same region”

*Also, while the collections of Panama are divided to three distinct locations the collections, in what I consider a larger area, Papua New Guinea is treated as one location. If one merges the diversities from Panama it gives the same percentage as PNG. Could the authors comment on this point, which echoes with the above recommendation to tone down conclusions about geographical specificity?*

We appreciate and generally agree with this comment, but would like to emphasize the following considerations. The Panama region is divided into 4 major parts, two on the Pacific Ocean (PAC/PAG) and two on the Caribbean Sea (PAB/PAP), representing very different geographical environments (e.g. different oceans). The sampling density and repetitive collections from the same locations in different years was much higher for the Panama samples than for the PNG samples. Additionally, the metadata for the Panama region was much better documented and provided us with higher spatial resolution than for the PNG area. Taking these considerations together, we would like to keep the current location categories as presented.

*3) Another concern is related to the selection of organisms includes (quite arbitrary) benthic filamentous marine cyanobacteria and various classes of macro-algae (Rhodophyta, Chlorophyta and Phaeophyceae) predominantly collected from warm waters (Carribean, Indo-Pacific, Indian Ocean…). It is nice to see but no surprise that MS/MS features for marine cyanobacteria and algae are different from marine and terrestrial actinobacteria, lichens, freshwater cyanobacteria and corals. In relation with the previous points on geographic specificity, the authors should more explicitly acknowledge this biological bias in the text.*

We thank the reviewer for highlighting this point. However, the difference in features as expected or “non-surprising” as they are, were never shown before, and if so, were not possible for comparison at this scale before the GNPS platform. The purpose of this study was to discover additional information from a well-studied collection of marine cyanobacteria and algae that have already yielded many new compounds for drug discovery efforts. The various classes of cyanobacteria and algae were field identifications from samples collected over the past 30 years and did not reveal any distinct classification. Also, it should be taken under account that these are environmental samples that harbor complex communities. Although, there are biases that may come from every step along the collection and analysis process, and this is something that needs to be kept in mind. Following this comment, we have *emphasized* this further in the text:

“However, not only is PNG a biodiversity hotspot, molecular networking quantitatively revealed that this region is rich in the presence of unique natural products compared to other locations sampled in this study”.

“As we can see in this dataset, presented here for illustrative purposes of the tools available in GNPS, most molecules are broadly distributed in disparate physical locations, whereas others are highly specific to particular locations.”

*4) Along the same line, the authors state that "86.3% were features that were unique to these collections." In some way, the authors should provide a better method of capturing the current state of GNPS to provide a benchmark. While Figure 1 provides the number of molecular features, it would be useful to have a better understanding of the number of molecules present in GNPS at the time the comparison was made. Perhaps, this information is captured in Supplementary file 4. Looking at Figure 1, it appears that most of GNPS might be comprised of data from cyanobacteria. How many of those features are represented in the current study? This information will be important for the reader to understand what the comparison entails.*

This is an important point, since by using GNPS, the data is kept “alive” and we are happy to address this comment:

At the time the comparison was made, GNPS held thousands of library standards from: FDA (>2300 spectra) NIH ((>3000) spectra from clinical collections, natural products library and active small molecules), Faulkner natural products legacy library (>120 spectra), Prestwick Phytochemical library (140 spectra), Massbank (>10000 spectra), HMBD (>2200 spectra), licensed NIST 2014 (for our in-house use) and the GNPS user contributions (>1500 spectra). In addition, our dataset is publicly available and undergoing continuous identification, meaning that every once in a while, all available datasets are reannotated against the updated library; hence additional annotations appear over time. We encourage scientists that are interested in our dataset (or any other) to subscribe to GNPS and get email updates once additional standards are annotated.

The number of annotated spectra and molecular families in this dataset at the time this work was processed is presented in the last paragraph of the subsection “Comparative metabolomics and chemical diversity of large scale datasets”.

We refer readers to the full dataset ([Supplementary-material SD2-data]) that is open for everyone to “rerun” against the updating library.

[Editors' note: further revisions were requested prior to acceptance, as described below.]

*1) First, the Abstract still conveys the general idea that the authors conducted a comprehensive regio-specific analysis while only 8 sampling sites were considered. As discussed in the first round of review, this is a technically understandable shortcoming which does not alter the validity of the MS/MS and data analysis approaches, but the fact that the study is limited to 8 sampling sites should be stated in the Abstract.*

We appreciate the editors comment and followed it, to state the fact that 8 major locations are presented. However, in order stay within the restricted word limit of the Abstract we removed the description of the compared datasets.

“Remarkably, 86% of the metabolomics signals detected were not found in other available datasets of similar nature, supporting the hypothesis that marine cyanobacteria and algae possess distinctive metabolomes. The data were plotted onto a world map representing 8 major sampling sites, and revealed potential geographic locations with high chemical diversity.”

*2) The lower panel of Figure 2 which is referred to as panel C in the third paragraph of the subsection “Comparative metabolomics and chemical diversity of large scale datasets” is not labelled accordingly in the figure. This minor labelling discrepancy should be corrected.*

Thank you for this comment. We have corrected accordingly, and referenced to the panels of Figure 2 only.

*3) Point 3 raised during the review concerned the interpretation of the comparison of features derived from Panama and Papua New Guinea (PNG) samplings to the overall chemical diversity across sampling sites. Most notably, if one merges the explained chemical diversities from Panama, it results in the same percentage as PNG. Considering the sampling diversity and existing metadata, the authors in their response to this point do not agree with the suggestion of merging the chemical diversity analyses inferred for each of the Panama current location categories. This argument for not merging the Panama location categories is reasonable; however, we feel that it remains inappropriate to draw the particular strong conclusion: "these numbers suggest that even though the sample sizes were similar, the detected chemical diversity varies greatly with PNG possessing the greatest chemical richness". This final sentence of the paragraph could be removed without altering the overall description of the value of the analytical approach used to explain sampling site-specific chemical diversity.*

We truly thank the editors for highlighting the issue and for the suggested solution. We have followed this comment and removed the sentence accordingly.

*4) Finally, since the term "digitizing" or "digitization" is used in the title as well as throughout the text as a cornerstone step in the authors' workflow, it may be helpful for readers that one or two sentences are added to precisely define this term from a technical standpoint. Readers from different backgrounds will have different expectations as to what is meant by "digitizing".*

We thank the editors for this comment and agree that this might be interoperate by readers in various ways. We have added a short explanation in parenthesis:

“In this study, we used a mass spectrometry based approach to digitize (convert to data format that can be stored, shared and analyzed by computational tools) the chemical inventory of an established marine cyanobacteria and algae collection in order to better evaluate its diversity and probe for novel natural products.”

One additional error was spotted and corrected. The text referred to Figure 1 and was corrected to 1C.